# Inter- and Intra-Observer Variations in Radiographic Evaluation of Pelvic Limbs in Yorkshire Terriers with Cranial Cruciate Ligament Rupture and Patellar Luxation

**DOI:** 10.3390/vetsci9040179

**Published:** 2022-04-10

**Authors:** Karol Ševčík, Marián Hluchý, Marieta Ševčíková, Michal Domaniža, Valent Ledecký

**Affiliations:** Small Animal Clinic, University of Veterinary Medicine and Pharmacy in Košice, Komenského 73, 041 81 Košice, Slovakia; sevcik.karol@gmail.com (K.Š.); marietakurillova@gmail.com (M.Š.); domaniza.michal@gmail.com (M.D.); valent.ledecky@uvlf.sk (V.L.)

**Keywords:** Yorkshire Terrier, cranial cruciate ligament, patellar luxation, hindlimb alignment, radiography

## Abstract

The main aims of the study were to describe bone alignment differences in Yorkshire Terriers (YT) with cranial cruciate ligament rupture (CCLR), patellar luxation (PL), or with a combination of both (CCLR + PL); to verify the theory of increased strain on cranial cruciate ligament (CrCL) due to PL as a predisposing factor; and to evaluate intra- and inter-observer variability of the protocols developed for measurement of femoral and tibial alignment in Yorkshire Terriers. Fifty-five hindlimbs of YT were divided into four groups: Control, CCLR, PL, and CCLR + PL. Thirty parameters were radiographically evaluated including hip joint, femoral, tibial, and intercondylar fossa (ICF) parameters. Three observers evaluated all parameters on two separate occasions with a twelve-week interval between measurements. Significant differences in conjunction with CCLR and PL theories between groups were noticed in: Norberg angle (NA), quadriceps angle (Qa), anatomic lateral distal femoral angle (aLDFA), femoral varus (FVA), mechanical cranial proximal tibial angle (mCrPTA), mechanical caudal proximal tibial angle (mCdPTA), tibial plateau angle (TPA), distal tibial axis/proximal tibial axis angle (DPA). Some interesting findings are the similarity of values between Control and CCLR vs. CCLR + PL and PL groups in Na, Qa, aLDFA; between CCLR + PL and PL in FVA and a significantly lower age of dogs in CCLR + PL compared to CCLR group. Based on our results, we can conclude that YT with different clinical findings have differing bone morphology. Moreover, these findings may support PL as a predisposing factor for CCLR in small breeds. Measurements in which excellent inter- observer agreement was achieved may be used for surgical planning or for further discussions.

## 1. Introduction

Cranial cruciate ligament rupture (CCLR) and medial patellar luxation (MPL) are two common causes of dysfunction and lameness in dogs. Generally, they are independent, but they often co-occur and mainly affect small breeds of dogs [1,2,3]. Co-occurrence of CCLR and MPL ranges generally from 6% to 25% [2,4].

Malalignment of the hindlimb bones, including coxa vara, genu varum, retroversion of the femoral head and neck, distal femoral varus, hypoplastic medial condyles, medial torsion of the tibia, proximal tibial valgus, and medial displacement of the tibial tuberosity have been reported as the most important predisposing factors for MPL [5]. Many causes of CCLR have been investigated, and common etiopathogeneses include age-associated degeneration of the ligament, autoimmune components, micro trauma, conformational abnormalities, genetic factors, and processes associated with breed and sex [6,7,8,9,10]. Suggested pathogenesis for dogs with MPL that develop concomitant CCLR is an increase in strain on the ligament as a result of anatomic abnormalities associated with MPL [5,11,12]. Conversely, investigators have hypothesized that dogs with CCLR with no previous history of an MPL, or with Grade 1 MPL and no clinical signs, may acquire an MPL, or increase clinical signs as a result of the increased internal rotation of the tibia once the cranial cruciate ligament has ruptured [5,11]. Patellar luxation (PL), medial or lateral, has also been reported as an unusual postoperative complication of CCLR stabilization surgery. The proposed mechanism for postoperative PL included a failure of incisional closure or tearing of the retinacular incision, potentially combined with muscle atrophy with a resulting lack of muscular control [11].

Some of conformational deformities, which may contribute to MPL, may subsequently alter the cranial cruciate ligament (CrCL), which in turn triggers a cascade of processes leading to ligament rupture [13,14]. The evaluation of hindlimb alignment has long been a discussed topic in veterinary orthopedics. Reference values would help to specify the quantitative degree of malalignment, particularly in the cases of bilateral affected limbs, in which a contralateral limb cannot be used as a reference one. The physiologic values may vary in different breeds, and therefore evaluation of normal bone alignment in different breeds is essential. Accurate determination of the magnitude of conformational deformity is crucial during surgical planning. Assessment of angular limb deformity in dogs is mostly performed via radiography and computer tomography (CT) [14,15]. Several studies have evaluated the repeatability and reproducibility of different radiographic protocols [15,16]. However, this method requires proper positioning and if malpositioning can be excluded, evaluation of radiographs is highly subjective and therefore, in such types of studies, there should be more than one observer.

Previous reports evaluating a higher number of parameters in small breeds were focused predominantly on dogs with PL [17]. There has not been a published comparative study aimed at morphological alignment of pelvic limbs with CCLR dogs, PL dogs, and with combination of both evaluated by three observers on two separate occasions yet. Therefore, the first aim of this study was to provide specific quantitative data of femoral and tibial conformation in Yorkshire Terriers divided into four groups. In conjunction with this part, we hypothesized that there would be significant differences in some measured parameters between the control group and affected groups and between affected groups and each other, with a specific focus on CCLR group vs. PL group.

Our second aim was to verify the theory of increased strain on CrCL due to PL, because, based on our knowledge, there has not been a study evaluating CCLR, PL, and the combination of both simultaneously in one specific breed.

The final purpose of this study was to evaluate intra- and inter-observer variability of the protocols developed for measurement of femoral and tibial alignment in Yorkshire Terriers.

## 2. Materials and Methods

We prospectively evaluated the hindlimbs of Yorkshire Terriers presented to the Small Animals Clinic of the University of Veterinary Medicine and Pharmacy in Košice (Slovakia), between October 2018 and December 2020. They presented with hind limb lameness localized in the stifle joint caused by cranial cruciate ligament rupture, patellar luxation, or both of them. All owners of the dogs used in this study agreed and signed a consent form of data collection. All dogs underwent orthopedic examination performed according to the standardized orthopedic protocol [5]. Inclusion criteria for the control group were: minimum age 12 years, bilaterally healthy, without orthopedic problems, and no previous hind limb surgery. Dogs for the control group were under general anesthesia for the reason unrelated to the study (dental care). Contralateral limbs were automatically excluded from the study. In the control group, the exclusion of the second limb was based on an assumption of similarity of values between limbs and therefore the elimination of subsequent artificial increasing of the number of patients included in the study was achieved. In the patients’ groups (CrCLr; CrCLr + PL; PL), only the affected limb was included in the study. In bilaterally affected dogs, the limb, which was the source of clinical issues of the dog, was evaluated. If the source of limping was bilateral, dogs were excluded from the study (during the study period, just two such dogs were presented).

### 2.1. Radiographic Methods

All radiographs were carried out on anesthetised dogs, using butorphanol (Butomidor, Richter Farma AG, Wels, Austria) (0.2 mg/kg), medetomidine (Cepetor, CP-Pharma MbH, Burgdorf, Germany) (10 μg/kg), and propofol (Propofol, Fresenius Kabi Deutschland GmbH, Bad Homburg, Germany) (2–4 mg/kg). They were obtained using a standard clinical X-ray unit (Gierth HF 200A, X-ray apparatus, GIERTH GmbH, Riesa, Germany) digitized with a computed radiography system (FCR Prima T2, CR-IR 392, Computed Radiography, Fujifilm Co., Tokyo, Japan) and saved as DICOM files.

A series of the following radiographs were taken for each dog:

Standard ventrodorsal hip extended radiographs were performed with inclusion criteria- symmetry of the pelvis and size and shape of obturator foramina, parallel femurs, tip of the lesser trochanter was visible at the medial aspect of the femur, the fabellae bisected by their respective femoral cortex [18] Two types of medio-lateral views were taken: stifle extended (135 ± 5°), where the extension was assessed by means of ‘eminence landmarks’ [19], and stifle and hock joint flexed at approximately 90°, according to previous criteria [20]. All radiographs that did not match an angle of 135 ± 5° were repeated until a correct angle was achieved. The craniocaudal view of each femur was performed with the dog in a sitting position, extended hip and femur parallel to the radiographic cassette, with the beam centered at the middle part of the diaphysis of the bone. Inclusion criteria were—tip of the lesser trochanter visible at the medial aspect of the femur, the fabellae bisected by their respective femoral cortex [21]. Caudocranial radiographs of the tibia were taken with the medial aspect of the calcaneus aligned with the intermediate tibial ridge [22]. For evaluation of intercondylar fossa (ICF), radiographs were taken with the beam directed approximately proximocaudally to distocranially at an angle of 12° from the femoral shaft and oblique 7° to parallel the ICF [23]. Dogs were positioned in sternal recumbency, the left hand of the assistant lifted the dog under the abdomen, the right hand pulled out and fixed the examined limb, and the other person set the required angles using a transparent goniometer. Imaging was repeated until the ideal ICF position was achieved.

### 2.2. Radiographic Measurements

The Norberg angle (Na) was measured as previously reported [18].

From a sitting position, craniocaudal radiographs of the femur were measured: femoral inclination angle (FIA) with symmetrical axis-based method (SYMAX) [24]; anatomic lateral proximal femoral angle (aLPFA); anatomic lateral distal femoral angle (aLDFA); mechanical lateral proximal femoral angle (mLPFA) and mechanical lateral distal femoral angle (mLDFA) [25,26]; femoral varus (FVA) [27] and quadriceps angle (QA) [28].

Femoral length (FL) was from mediolateral radiographs as previously described [15].

Mediolateral 90°–90° radiographs were used for evaluating tibial plateau angle (TPA) [20]; relative tibial tuberosity width (rTTW) [29]; anatomical-mechanical axis angle (AMA angle) [30,31]; mechanical cranial proximal tibial angle (mCrPTA); mechanical caudal proximal tibial angle (mCdPTA); mechanical cranial distal tibial angle (mCrDTA); mechanical caudal distal tibial angle (mCdDTA); distal tibial axis/proximal tibial axis angle (DPA) [25,32,33]; tibial length (TL); proximal tibial width (PTW); distal tibial width (DTW); femoral condylar length (FCL); femoral width (FW) [34]; and Z angle [35].

The caudocranial view of the tibia was used for measuring the mechanical medial proximal tibial angle (mMPTA) and mechanical medial distal tibial angle (mMDTA) [22,25].

The ICF width was measured as cranial (A), central (B), and caudal (C) (Figure 1). The ICF height was measured from the apex of the ICF to a line connecting the distal surfaces of the femoral condyles (E). Total condylar width was measured from the lateral to medial epicondyles at the widest point (D) [23], and height of femoral condyles (F) was defined as the length of a line perpendicular to 2 lines, each parallel to the cranial cortex of the femur and located along the distal aspect of the femoral condyles and the proximal trochlear ridges of the femur [34]. After measuring these values, notch width indexes (NWI) were calculated: cranial notch width index (CrNWI) as A/D, the central NWI (CNWI) as B/D, caudal NWI (CaNWI) as C/D, and notch shape index (NSI) as B/E. Intercondylar notch height index (HICN) was calculated as E/F [13,23].

### 2.3. Statistical Analysis

Body weight, gender, and reproductive status data were collected. Measured values were grouped into four groups: control limbs; limbs with non-traumatic CCLR; limbs with CCLR + PL; PL limbs. Data were analyzed using one-way ANOVA with post-hoc Tukey’s test. Differences were considered significant at *p* < 0.05. Results were expressed as mean ± standard deviation.

Intra- and inter-observer agreement was evaluated with the two-way random single measures intra-class correlation coefficient for absolute agreement (ICC 2,1) [36]. Measurements were grouped separately for inter-observer agreement, and for intra-observer, they were grouped as first and second round. The ICC ranged from 0 (no agreement) to 1 (perfect agreement). ICC < 0.5 was considered as poor reliability, values between 0.5 and 0.75 indicated moderate reliability, values between 0.75 and 0.9 indicated good reliability, and values greater than 0.90 indicated excellent reliability. There was a twelve-week interval between the first and second rounds of measurements for all three observers.

Linear regression was used to compare DPA and TPA values.

Statistical analysis was performed using IBM SPSS version 27 statistical software and Graph Pad Prism 7.0.

## 3. Results

### Animals

Fifty-five hindlimbs of 55 Yorkshire Terriers were evaluated in this study, involving 32 females, or 59% (18 spayed, 56%), and 23 males, or 41% (14 castrated, 61%). The (1) CCLR group consisted of 14 limbs of dogs with mean body weight 5.1 kg (2.5–8.5 kg) and mean age 9 years (7.5–10.5 years); the (2) PL group included 14 limbs belonging to dogs with mean body weight 3.9 kg (2.5–6.5 kg) and mean age 4.8 years (2–7 years). The (3) CCLR + PL group comprised 15 limbs belonging to dogs with mean body weight 3.8 kg (2.4–5.8 kg) and mean age 7.1 years (4–10 years). The (4) control group was made up of 12 limbs belonging to dogs with mean body weight 4.9 kg and mean age 13.2 years; the ten stifles in the CCLR + PL group had grade 2 PL and five had grade 3 PL. In the PL group, 11 stifles had grade 2 PL and 3 stifles had grade 3 PL. Only three dogs, during the study, were found to have the grade 4 PL and four dogs had the grade 1 PL and therefore were excluded from the study due to their low number. There was a significant difference between all groups in age (*p* < 0.05) (Figure 2). The mean values, standard deviations, and statistical significance between groups are shown in Table 1.

To assess the inter-observer reliability of measurements, the agreement between three observers using the same methods was determined. Excellent ICC was detected in 14 (47%) measurements, good ICC in seven (23%) measurements, moderate in six (20%) measurements, and poor ICC in three (10%) measurements. Two observers were younger orthopedic assistants and the third was a younger assistant at the diagnostic imaging. All observers were dealing with orthopedic measurements on a daily basis.

To assess the intra-observer reliability of measurements, the agreement between the two repeated measurements by each observer was examined. ICC of the first observer was excellent for 10 (33%) measurements, good for 16 (53%), moderate for three (10%), and poor for one (3%) measurement. ICC of the second observer was excellent for three (10%) measurements, good for nine (30%), moderate for 11 (37%), and poor for six (20%) and one index had a negative value of −0.03 (HICN) (3%). ICC of the third observer was excellent for one (3%) measurement, good for 17 (57%), moderate for 11 (37%), and poor for one (3%) measurement. ICC for intra- and inter-observer results are given in Table 2.

## 4. Discussion

Yorkshire Terriers were selected for investigation in this study because they are frequently admitted to our clinic with CCLR, PL, or a combination of both. To decrease the variability associated with anatomical differences among breeds, it is generally preferable to focus on a single breed or several breeds but with a sufficient number of dogs. Some studies evaluated a combination of different breeds and the results were reported as an average, which could lead to incorrect surgical planning [37,38,39,40]. Based on Aghapour et al. [17], the majority of recent systematic reviews of similar small breed studies were focused on MPL and only few investigated CrCL disease, of which most of them have been discussing tibial sagittal plane measurements. To the best of our knowledge, there have not been any studies evaluating the inter- and intra-observer variability in hindlimb alignment measurements comprehensively in small-breed dogs with CCLR, CCLR + PL, and PL.

First of all, we found a significant increase in age from the PL group, through CCLR + PL to the CCLR group, which is in line with previous findings where dogs with PL had a significantly lower age compared to the CCLR + PL group of dogs [2,41]. It was previously theorized that increased strain on CrCL might be a consequence of PL [1,5]. The greater age of dogs in the CCLR compared to CCLR + PL group in our study may support this statement and therefore, we agree with the suggestion that YT with PL have a predisposition for CCLR [2,37]. It would be interesting to determine the prevalence of CCLR in small-breed dogs with already surgically corrected PL.

To the present authors’ knowledge, there are no published data concerning the physiological Norberg angle values for YT. Our results show significant differences between the control group vs. CCLR + PL and PL groups, but not between the control group vs. CCLR. Due to the similarity of the measured values between CCLR + PL and PL group, and on the other hand, differences with CCLR, we may consider PL as one of the contributing factors for CCLR development.

We decided to use the SYMAX method for FIA measurement. Compared to other methods, SYMAX measurements were most consistent and therefore more suitable for evaluation of our observers’ measurements [42]. To discuss FIA values, we selected only small breed studies and those focused on one specific breed; unfortunately, all dealt with PL and not CCLR [43,44,45,46]. The study focused on large-breed dogs’ described difference between FIA values of CCLR dogs against control ones [15]. The values gained by us has shown non-significant differences between groups and are comparable or slightly higher than in previously reported studies focusing on PL [43,44,45,46], which suggests that coxa vara is not associated with either PL or CCLR. The plausibility of our results is confirmed by excellent or good inter- and intra-observer agreement, except for the second observer, who achieved moderate intra- agreement for FIA.

Radiographically measured QA angle is a highly subjective parameter because the origin of the rectus femoris muscle is not radiologically visible, and difficulty in identifying the tibial tuberosity may be the source of significant differences in observers’ measurements or between studies. Paradoxically, the inter-observer agreement for QA was good in this study, probably because the two observers do not perform this measurement regularly, and therefore, they undertook practice measurements on radiographs not included in the study. Intra-observer agreement was excellent for the first observer, good for the second, and moderate for the third. We found significant differences between the Control vs. CCLR + PL and PL groups; and between CCLR vs. CCLR + PL and PL groups. Mean values in the Control and CCLR groups was 15.4° and 15.6°, respectively, which is higher than in large breeds [15,47]. Although it has recently been reported that high QA (18.3°, 80 dogs) in small-breed dogs without PL may be an objective parameter for explaining the major biomechanical predisposition of small dogs to PL [47]. Similarity of values between Control and CCLR against CCLR + PL and PL might again slightly favor PL as the factor contributing to CrCL alteration.

It cannot be denied that incidence of femoral deformities in the distal part of the bone is greater than in the proximal portion, and evaluation of aLDFA, mLDFA, and FVA is appropriate. According to the published values of aLDFA, mLDFA, and FVA, which focused on one specific breed, the most pronounced difference was found in dogs with grade 4 PL [37,44,45,46], which we cannot compare because grade 4 PL was not evaluated in this study, compared to lower grades [37,44,45]. On the other hand, our values of the Control and PL group are in line with those previously reported [43,44,45,46,48]. Moreover, if we look carefully at aLDFA values, there is obvious similarity between Control and CCLR vs. CCLR + PL and PL values, which led to the theorization about PL as the predictor factor of CCLR again. Likewise, are FVA values of CCLR + PL and PL. The strength of these findings is supported by excellent and good inter- and intra-observer agreement, so the results appear to be highly reproducible.

We evaluated seven parameters of the proximal tibia in the sagittal plane, which may affect the stifle. It has been reported that caudal proximal tibial deformity, mostly in small-breed terriers, may expose the CrCL to increased risk [49], which is consistent with our findings of significantly lower mCdPTA in the CCLR group compared to CCLR + PL and PL. CCLR + PL and PL groups did not differ in this respect, which may again slightly support the theory of increasing stress on the CrCL due to PL. A large number of studies have evaluated TPA, and our findings of significantly higher TPA in CCLR group compared to the others are in line with them [33,38,50]. It is very likely that malalignment between the anatomical and mechanical axes of tibia may be induced as a consequence of the caudal angulation of the proximal tibia, which may result in a not-fully-aligned proximal anatomic axis with longitudinal anatomic axis [30]. The AMA angle is used to quantify caudal angulation of the proximal tibia [32,33] and recently has been suggested as a clinically relevant predisposing factor for the development of CrCL rupture in dogs [50]. However, our results are not significantly different between any of our group and failed to prove this statement. Nevertheless, there are not described AMA angle values of YT in literature yet. Therefore, they may serve as a source for continuous discussion about the AMA angle. RTTW and Z angle values did not show any significant difference between the groups and appear to be slightly higher than those previously reported [51]. Z angle can be considered as a marker of proximodistal tibial tuberosity position, which may help to decide which tibial osteotomy is appropriate for dogs with CCLR if the reference values are known. It would be interesting to try to define reference values of the Z angle for osteotomy selection. The potential effects of rTTW and Z angle on surgery planning are already well discussed [51]. All these parameters (including mCrPTA and PTW) had an excellent or good inter- and intra-observer agreement in our study, apart from the intra-observer agreement for the second observer regarding rTTW and Z angle, whose values were moderate but slightly below the limit.

The term ‘proximal shaft deformity’ identifies dogs with DPA greater than 11.23°, and it has been hypothesized that this contributes to increased TPA and may represent a risk factor for CCLR [33]. Direct correlation between TPA and DPA has been demonstrated in medium to large dog breeds [33]. Our results support these findings because both DPA and TPA values in CCLR were significantly higher against the control group. Linear regression has shown moderate correlation of DPA and TPA values (R = 0.51). Furthermore, DPA measurement has been suggested as a facilitating tool for optimization of the surgical management of CCLR dogs [33].

Computed tomography (CT) provides many advantages in terms of allowing the ICF evaluation, and the adequacy of radiography has been questioned. However, with careful attention to positioning of the limbs, radiographs can provide accurate evaluation of the ICF and there may be no significant differences between radiographic, CT, and gross evaluation of ICF [52]. Additionally, it was determined that if the gantry angle is either under- or over-rotated by up to 4° from the ideal 12° off the cranial/dorsal aspect of the femur, then the measurements taken within the intercondylar notch will not be significantly affected [53]. Comerford et al. reported significantly lower NWIs (cranial, central, and caudal) in breeds predisposed to CCLR compared to low-risk breeds [13]. This finding was confirmed by a comparison of CCLR stifles and healthy ones [54]. More significant differences were observed in CrNWI and CNWI [13], which is in line with our findings with significantly smaller CrNWI and CNWI in CCLR YT compared to the others. Although we also found differences among the other groups, including CNWI and CaNWI measurements, it is difficult and questionable to draw conclusions from them, given that those groups had PL, which is associated with femoral deformities and their impact on ICF alignment and measurements is unknown. Inter-observer agreement was moderate in four out of six ICF parameters, while two parameters had poor agreement (NSI and HICN). None of our observers performed these measurements routinely, they even practiced them before the study, and their results are lower compared to the other measured parameters. Intra-observer agreement ranged from negative (one parameter for one observer), through zero (also one parameter for same observer) to good, according to the scale we used. The results from this study suggest that radiographic evaluation of ICF has low reproducibility, and this should be kept in mind if someone would like to compare such results.

We used ICC for evaluation of observer reliability, and interpretation of ICC values is a non-trivial task. There are different forms of this test which can produce different results even in evaluating the same values, and the results may give the impression of high agreement. The precise definition of the statistical method used should be standard in research reports. In studies where the statistical method is not precisely defined, it is impossible to compare the results and, in addition, there are different evaluation scales for ICC; a more critical one was used in this study. Another limitation of similar studies is an undefined or non-uniform interval between intra-observer measurements, which can also affect the measurements, their reliability, and repeatability.

There are two major limitations to this study: the relatively low number of dogs included and the absence of grade 4 PL. On the other hand, number of limbs included in this study is presumably not as big of a limitation as could be thought because we had similar or higher numbers compared to the others [37,43,44,45,48]. We do not think the absence of grade 1 PL is a major limitation because previous results did not show severe differences between control and grade 1 PL groups based on a recently published systematic review [17]. In our experience, it is routinely diagnosed grade 2 and 3 PL, whilst it is not common to observe grade 4 PL. Therefore, we were not able to compare our results with those included in grade 4 PL. Due to the insufficient number of dogs with grade 2 and grade 3 PL, the CCLR + PL and PL are composed of limbs combination of both; even the majority of limbs were diagnosed with grade 2 PL, and therefore the values are probably more suitable for grade 2 PL.

## 5. Conclusions

The main purpose of this study was to define hindlimb alignment values of YT with various clinical findings. Based on our results, we can conclude that YT with different clinical findings have differing bone morphology. Our results may be a part of a great process of collecting values from different breeds with different clinical findings and may help surgeons with appropriate surgical planning if needed.

Based on increasing age from PL through CCLR + PL until CCLR and measured differences in NA, Qa, aLDFA, mCdPTA, may indicate PL as a predisposing factor for CCLR development. These results might help with understanding of the CCLR pathophysiology in a certain type of breed.

Two thirds of the inter-observer agreements were excellent or good. Therefore, we can conclude that majority of parameters are highly reproducible. On the other hand, ICF measurements have shown insufficient reproducibility.

## Figures and Tables

**Figure 1 vetsci-09-00179-f001:**
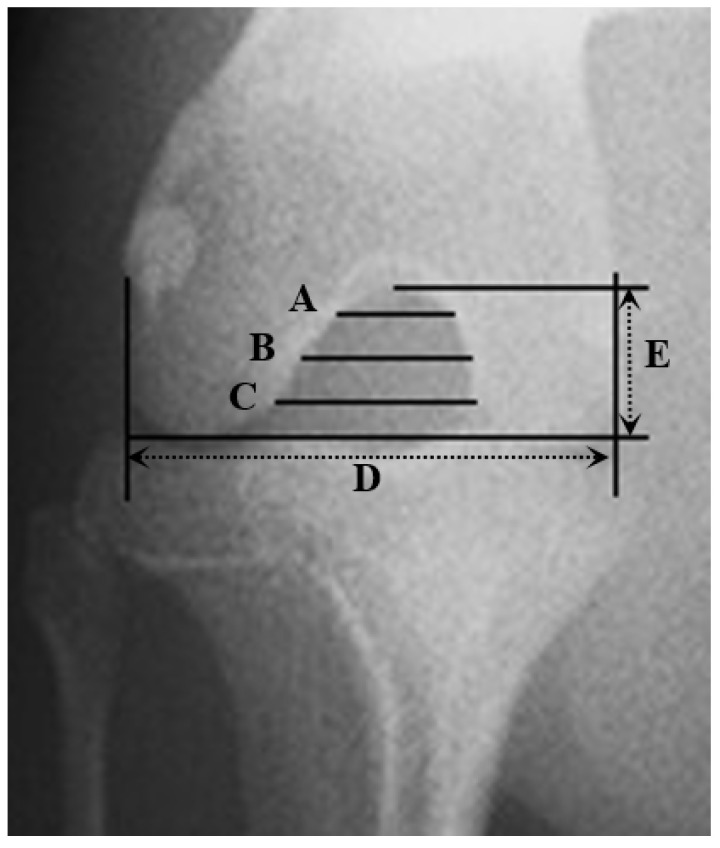
Radiographic measurement of ICF. Cranial (**A**), central (**B**), and caudal (**C**) ICF width, (**D**)—total condylar width, (**E**)—ICF height.

**Figure 2 vetsci-09-00179-f002:**
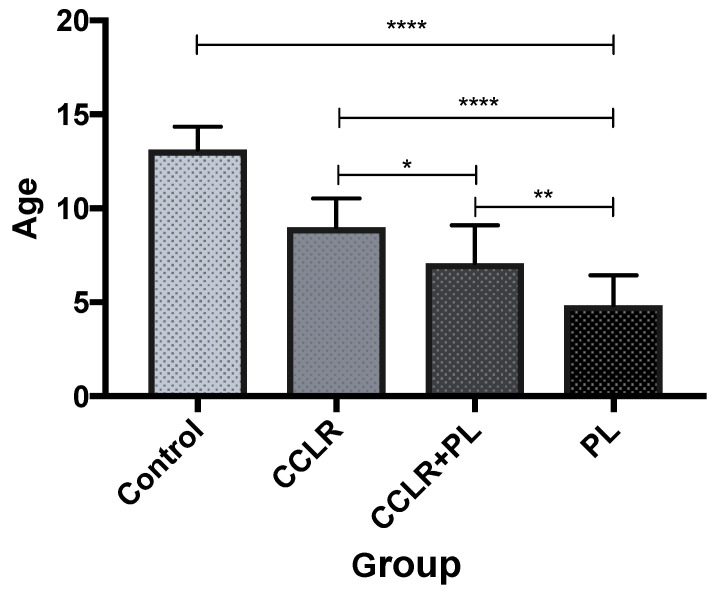
Differences between groups of dogs in age. * *p* < 0.05, ** *p* < 0.01, **** *p* < 0.0001.

**Table 1 vetsci-09-00179-t001:** Hindlimb parameters (mean ± SD) and statistical significance (*p* < 0.05) of YT categorized into four groups.

Group	NA _b c_	FIA	Qa _b c d e_	aLPFA	aLDFA _b c_	mLPFA	mLDFA	FVA _a b c d_	FL	mCrPTA _a d e_
**Control**	107.2 ± 3.3	132.7 ± 2.9	15.6 ± 2.2	120 ± 3.3	96.1 ± 2.2	116.6 ± 4.2	101.1 ± 2.7	5.8 ± 1.9	71 ± 6.7	118.9 ± 3.4
**CCLR**	101.7 ± 4.6	132.4 ± 5.3	15.4 ± 5.1 a b	121.6 ± 8.6	98.5 ± 2.4	116.5 ± 7.8	102.6 ± 2.3	8.8 ± 2.5	71 ± 3.6	123 ± 4.5
**CCLR + PL**	96.3 ± 9.9	133.9 ± 5.9	22.7 ± 4.6	122 ± 6.3	100.6 ± 3.9	114.8 ± 7.4	101.7 ± 15.6	11.8 ± 3.2	71.4 ± 6.3	116.6 ± 3.8
**PL**	98 ± 8.5	133.5 ± 4.9	20.5 ± 2.9	119.5 ± 6.9	100.2 ± 3.1	113.1 ± 5.1	103.6 ± 3.6	10.5 ± 3.5	74.2 ± 6.6	117.5 ± 3.9
	**mCdPTA** ** _d e_ **	**mCrDTA**	**mCdDTA** ** _e_ **	**mMPTA**	**mMDTA**	**TPA** ** _a d e_ **	**AMA**	**rTTW**	**Z angle**	**PTW** ** _d_ **
**Control**	60.9 ± 4	91.5 ± 2.4	87 ± 2.5	93.8 ± 3	92.6 ± 2.2	27.1 ± 3.1	4.6 ± 0.39	0.89 ± 0.05	74.2 ± 3.9	18 ± 2.3
**CCLR**	57.9 ± 1.9	92.6 ± 2.57	86.2 ± 2.6 b	92.2 ± 1.8	92.6 ± 2.6	32.6 ± 1.6	4.2 ± 0.51	0.91 ± 0.09	73.3 ± 2.4	19.6 ± 1.7
**CCLR + PL**	63.1 ± 4	92.1 ± 3.2	87.6 ± 3.4	94.4 ± 4.7	92.7 ± 3.5	28.6 ± 4.5	4.1 ± 1.9	0.86 ± 0.07	73.6 ± 4.1	17.7 ± 2
**PL**	62.8 ± 4.1	91.8 ± 3	88.8 ± 2.2	93.2 ± 4.2	92.2 ± 2.5	26.8 ± 3.7	3.44 ± 1.5	0.87 ± 0.05	72.2 ± 3.6	18.6 ± 1.9
	**DTW**	**TL**	**FCL**	**FW**	**DPA** **a**	**CrNWI** ** _a d e_ **	**CNWI** ** _a d_ **	**CaNWI** **f**	**NSI**	**HICN**
**Control**	5.3 ± 0.6	73.9 ± 6.5	16.9 ± 1.9	5.5 ± 0.75	7.6 ± 2.8	0.23 ± 0.01	0.31 ± 0.02	0.32 ± 0.02	0.94 ± 0.13	0.29 ± 0.02
**CCLR**	5.9 ± 0.7	74.8 ± 5.7	17.1 ± 1.4	5.7 ± 0.57	11.3 ± 4.8	0.17 ± 0.01	0.27 ± 0.03	0.32 ± 0.01	0.97 ± 0.03	0.29 ± 0.01
**CCLR + PL**	5.4 ± 0.7	72.3 ± 5.6	16.3 ± 1.6	5.4 ± 0.71	10 ± 2.4	0.22 ± 0.02	0.30 ± 0.01	0.33 ± 0.03	0.85 ± 0.33	0.3 ± 0.02
**PL**	5.6 ± 0.9	74.2 ± 6.5	16.6 ± 1.7	5.8 ± 0.85	8.8 ± 4.3	0.21 ± 0.02	0.29 ± 0.02	0.31 ± 0.03	0.78 ± 0.33	0.3 ± 0.01

_a, b, c—_significant difference between Control and CCLR _a_, CCLR + PL _b_, PL _c_ group; _d_—significant difference between CCLR and CCLR + PL groups; _e_—significant difference between CCLR and PL groups; _f-_ significant difference between CCLR + PL and PL. NA—Norberg angle, FIA—femoral inclination angle, QA—quadriceps angle, aLPFA—anatomic lateral proximal femoral angle, aLDFA—anatomic lateral distal femoral angle, mLPFA—mechanical lateral proximal femoral angle, mLDFA—mechanical lateral distal femoral angle, FVA—femoral varus angle, FL—femoral length, mCrPTA—mechanical cranial proximal tibial angle, mCdPTA—mechanical caudal proximal tibial angle, mCrDTA—mechanical cranial distal tibial angle, mCdDTA—mechanical caudal distal tibial angle, mMPTA—mechanical medial proximal tibial angle, mMDTA—mechanical medial distal tibial angle, TPA—tibial plateau angle, AMA—anatomical-mechanical axis angle, rTTW—relative tibial tuberosity width, Z—angle, PTW—proximal tibial width, DTW—distal tibial width, TL—tibial length, FCL—femoral condylar length, FW—femoral width, DPA—distal tibial axis/proximal tibial axis angle, CrNWI—cranial notch width index, CNWI—central notch width index, CaNWI—caudal notch width index, NSI—notch shape index, HICN—intercondylar notch height index.

**Table 2 vetsci-09-00179-t002:** Intra-class correlation coefficient for inter- and intra-observer agreement.

	ICC Inter	ICC Intra		
		1	2	3
**N. angle**	0.98	0.98	0.9	0.9
**FIA**	0.89	0.91	0.68	0.89
**Qa**	0.76	0.91	0.6	0.84
**aLPFA**	0.91	0.89	0.83	0.8
**aLDFA**	0.91	0.84	0.81	0.85
**mLPFA**	0.85	0.88	0.73	0.91
**mLDFA**	0.91	0.85	0.91	0.9
**FVA**	0.96	0.82	0.98	0.85
**FL**	0.93	0.92	0.87	0.83
**mMPTA**	0.92	0.73	0.74	0.74
**mMDTA**	0.91	0.74	0.67	0.73
**mCrPTA**	0.89	0.93	0.81	0.81
**mCdPTA**	0.89	0.95	0.78	0.85
**mCrDTA**	0.66	0.84	0.44	0.77
**mCdDTA**	0.63	0.82	0.4	0.71
**TPA**	0.97	0.94	0.83	0.83
**AMA**	0.94	0.9	0.88	0.81
**rTTW**	0.85	0.85	0.73	0.83
**Z angle**	0.89	0.83	0.71	0.77
**PTW**	0.92	0.92	0.91	0.87
**DTW**	0.92	0.89	0.73	0.7
**TL**	0.91	0.95	0.8	0.83
**FCL**	0.7	0.79	0.47	0.63
**FW**	0.96	0.91	0.74	0.73
**DPA**	0.46	0.79	0.33	0.64
**CrNWI**	0.56	0.76	0.58	0.7
**CNWI**	0.6	0.65	0	0.55
**CaNWI**	0.54	0.77	0.46	0.62
**NSI**	0.46	0.76	0.55	0.7
**HICN**	0.22	0.45	−0.03	0.3

## Data Availability

The data presented in this study are available on request from the corresponding author.

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
