# Peer review of "Inter- and Intra-Observer Variations in Radiographic Evaluation of Pelvic Limbs in Yorkshire Terriers with Cranial Cruciate Ligament Rupture and Patellar Luxation"

_vetsci, 2022, doi:10.3390/vetsci9040179_

Round 1

Reviewer 1 Report

Yorkshire terriers are dogs frequently involved in these joint pathologies, probably more so than other breeds because they have long bones deformed by artificial selection. The manuscript is interesting and sounds good. Well written and with detailed measurement tables. I suggest only few minor points in order to improve your manuscript:

  • Why the control group went to your clinic should be described.
  • The Figure 1 is not very beautiful. It could be improved and associated to other figures. Please also add a bar scale. 
  • Radiographical figures of control, patellar lussation (PL) and cranial cruciate ligament ropture (CCLR) should be shown.
  • Are you sure that in all your clinical cases there is not also a case of caudal cruciate ligement ropture?

Reviewer 2 Report

It is very interesting study study concerning bone alignment differences in Yorkshire Terriers  with cranial cruciate ligament rupture ,patellar luxation  or with combination of both above mention orthopedics problems. Additionaly authors  very deeply verify theory of increased strain on cranial cruciate ligament  due to patellar luxation as a predisposing factor. Methodology , statistical analysis were planed very properly. Exceptionally interesting is discussion section. In this chapter authors very thoroughly analized world literatures in the  presented area. Conclusion is a logical observation of performed  study and results that Yorkshire  Terriers with different 
clinical findings have differing bone morphology. Authors findings may support patallar luxation as a predisposing factor for cranial cruciate ligament rupture in small breeds. Justified is athors's observation that measurements in which excellent inter- observer agreement was achieved may be used for surgical planning or for further discussions. Article is worth to be publish in Veterinary Sciences.

Author Response

Thanks for reviewing the manuscript

Reviewer 3 Report

Dear authors,

please find below my suggestions to the  paper titled "Inter- and intra-observer variations in radiographic evaluation 2 of pelvic limbs in Yorkshire terriers with cranial cruciate ligament rupture and patellar luxation": 

Ln 13: delete “try”

Ln 11-27 Abstract needs to be revised according to the following suggestions

Ln 36: please list the bone abnormalities that lead to MPL in dogs

Ln 58-59:  other possibility of limb deformities assessment?

Ln 72: delete “is possible”

Ln 75-77: add this aim in the abstract too

Ln 79 must be clarified: 1) any ethical approval? 2) Owners signed a consent form?

Ln 83: please change: …all dogs underwent orthopaedic examination (in a different sentence);

Ln 79-83: inclusion criteria from pathological group need to be clarified

Ln 83-85: why did you have include just dogs older than 12 year in the control group?

Ln 86-87: “…contralateral limb also from the control group was excluded..”did you included just monolateral limb? Why? No Bilateral condition? Please clarify this sentence and add a comment in the results and discussion sections.

Ln 89: am quite concern performing an alignment X-rays study just with a sedation. Please clarify which protocol you have used.

Ln 105: axial femoral view looks missing. How did you check the FN anteroversion angle and femoral torsion?

Ln 114-119: please provide a figure with measurements

Ln 165: please list the groups like that: 1) CCLr; 2) PL; 3) CCLR+ PL; 4) control group

Ln 165: please add body weight and age ranges for each group

Ln 172-175: the PL grade is assessed during the ortho examination. Provide more information regarding the ortho exam (ln 82-83) and then report a separate paragraph in the results section with the clinical findings you observed. Any bilateral conditions?

Ln 194-197: please provide the level of experience of each observer. In the MM section there is no mention about the three observers. This is very relevant to know. Please add.

Ln 194-206: you scaled the ICC results without any mention in the MM. Please reference excellent, moderate, good, poor.

Ln 214: please add references

Ln 229-232: you opinion looks very ambiguous so better clarify otherwise delete this sentence. Please check ref 38 (Fluckiger et al I suppose)

Ln 233-234: differences regarding which parameters??

Ln 234-236: again, information regarding the femoral neck anteversion angle ans consequent femoral torsion are very relevant to support this sentence ; please state how you checked this measurement.

Ln 238: please use a new paragraph

Ln 258: I assume you are talking about the QA. So please do not use a new paragraph

Ln 266-276: please see above, comment ln 105.

Ln 283-284: please clarify what you means as ambiguous results.

Ln 458 -459 (ref. 37-38): please check

Round 2

Reviewer 3 Report

Dear Authors,

thank you for addressing my suggetions.

Please find below my comments

"Axial view is missing because of exclusion due to technical issues,Maybe this was not a good decision, but there are still enough parameters to characterise morphology".

This is not a good decision. You can avoid using the axial view if you can't, but you can easily estimante the femoral neck anterversion by goniometry. Please check  the Atlas by Petazzoni and Jaeger at pag. 54.

Please add the atlas in the references.

Author Response

Dear reviewer,

Thank you for quick reviewing.

As we have suggested in the first response we are capable of adding the measurement, if it is inevitable, as described by Petazzoni and Jaeger at page 54, it is a question of few days for inter-observer measurement. On the other hand, to follow our methodology, there would be required period at least three months for intra-observer measurements, because it was built on twelve week interval.

Based on this, your recommendation would be necessary, to ask editor for major revision, which would take slightly more than 3 months.